# Relationship between self-reported listening and communication difficulties and executive function: a protocol for a systematic review and meta-analysis

Jemaine E Stacey ![iD],[1] Henrik Danielsson,[2] Antje Heinrich,[3] Lucija Batinović,[2] Emil Holmer,[2] Elisabeth Ingo,[2] Helen Henshaw ![iD] [4,5]

For numbered affiliations see end of article.

**Correspondence to**
Dr Helen Henshaw;
helen.henshaw@nottingham.ac.uk

## ABSTRACT

**Introduction** Listening and communication difficulties can limit people's participation in activity and adversely affect their quality of life. Hearing, as well as listening and communication difficulties, can be measured either by using behavioural tests or self-report measures, and the outcomes are not always closely linked. The association between behaviourally measured and self-reported hearing is strong, whereas the association between behavioural and self-reported measures of listening and communication difficulties is much weaker, suggesting they assess different aspects of listening. While behavioural measures of listening and communication difficulties have been associated with poorer cognitive performance including executive functions, the same association has not always been shown for self-report measures. The objective of this systematic review and meta-analysis is to understand the relationship between executive function and self-reported listening and communication difficulties in adults with hearing loss, and where possible, potential covariates of age and pure-tone audiometric thresholds.

**Methods and analysis** Studies will be eligible for inclusion if they report data from both a self-report measure of listening difficulties and a behavioural measure of executive function. Eight databases are to be searched: MEDLINE (via Ovid SP), EMBASE (via Ovid SP), PsycINFO (via Ovid SP), ASSIA (via ProQuest), Cumulative Index to Nursing and Allied Health Literature or CINAHL (via EBSCO Host), Scopus, PubMed and Web of Science (Science and Social Science Citation Index). The JBI critical appraisal tool will be used to assess risk of bias for included studies. Results will be synthesised primarily using a meta-analysis, and where sufficient quantitative data are not available, a narrative synthesis will be carried out to describe key results.

**Ethics and dissemination** No ethical issues are foreseen. Data will be disseminated via academic publication and conference presentations. Findings may also be published in scientific newsletters and magazines.

**PROSPERO registration number** CRD42022293546.

## STRENGTHS AND LIMITATIONS OF THIS STUDY

⇒ This systematic review is the first to investigate the relationship between self-reported listening and communication difficulties and executive function using meta-analysis to synthesise the available evidence.

⇒ It uses an established framework (International Classification of Functioning and Disease) and taxonomy (Cattell-Horn-Carroll-Miyake) to define target domains and measures of listening and communication difficulties and executive functions, respectively.

⇒ Grey literature (including unpublished study results) will be included.

⇒ This protocol has been reported in accordance with the PRISMA-P statement.

⇒ Only studies available in English are eligible for inclusion.

can limit people's participation and adversely affect their quality of life. Hearing loss plays a fundamental role in determining a person's ability to listen and communicate, although research over the years has shown that there are other factors, beyond hearing loss, that are also important. Both hearing and listening and communication can be measured using behavioural measures such as pure-tone audiograms and speech-in-noise tests, or via self-report questionnaires. Behavioural and self-report measures of hearing are generally well correlated, and behavioural measures of hearing are often well correlated with both behavioural and self-report measures of listening and communication, that is, questionnaires. When a listener has a behaviourally measured hearing loss, it is likely that they will also experience and report difficulties with listening and communicating.

Effective communication, which relies on good hearing, is instrumental for a high level of functioning and good quality of life.[1]

## INTRODUCTION

Listening and communication are crucial for a healthy life and difficulties in communication

**Table 1** Frameworks describing hearing and effective communication

| Kiessling *et al*[2] | ICF framework | |
|---|---|---|
| | Body functions = physiological functions of body systems | |
| Hearing: a passive function that provides access to the auditory world via the perception of sound | b230 | Hearing functions |
| | Activities and participation = execution of a task or action by an individual and involvement in a life situation | |
| Listening: the process of hearing with intention and attention | d115 | Listening |
| Comprehending: the reception of information, meaning or intent | d310 | Communicating with – receiving – spoken messages |
| Communication: the bi-directional transfer of information, meaning or intent between two or more people | d350 | Conversation |

ICF, International Classification of Functioning and Disease.

Kiessling *et al* proposed a cascade linking hearing to effective communication, which in turn can be mapped to the International Classification of Functioning and Disease (ICF) framework (core set for hearing loss).[2] Both frameworks are displayed in table 1.

On the other hand, considering the activities and participation domain of listening and communication, behavioural and self-report measures are less robustly correlated. This is highlighted by the fact that two individuals who experience the same pure-tone average audiometric thresholds can experience and report substantially different degrees of listening and communication difficulties.[3] One interpretation of this result could be that they assess slightly different concepts and/or highlight different contributing factors. One of those contributing factors whose role still remains to be fully understood is cognition.[4]

Cognition has a complex relationship with auditory function depending on whether it is considered at the function (hearing) or activities/participation (listening/communication) level. Specifically, hearing loss, both behaviourally measured and self-reported, has been shown to be associated with poorer cognitive performance across a range of cognitive domains including global cognition, episodic memory, processing speed, semantic memory, visuospatial ability, executive functions, and cognitive impairment and dementia.[5] Indeed, Marrone and colleagues[6] reported that adults reporting any trouble hearing were at nearly four times higher odds

of reporting increased confusion and memory loss and half as likely to report good general health compared with adults reporting no hearing difficulty. These results are important to acknowledge because hearing loss has been identified as the leading potentially modifiable risk factor for dementia in midlife.[7]

For listening and communication, on the other hand, the type of assessment appears to play a role. For behavioural measures, the role of cognition for the ability to perceive speech (and in particular, speech in noise) has been reliably demonstrated for individuals with hearing loss, and this relationship is robust even when taking into consideration individuals' age and objective hearing levels (pure-tone average audiometric thresholds).[8] Note that the cognitive ability most commonly assessed in studies of speech perception in noise is working memory. Other abilities such as attention and executive function are less regularly assessed and less robustly found to link to speech perception in noise. One reason for the less robust link might be that the speech in noise perception task needs to be a particular type or of more complexity in order to necessitate attentional and executive functions.

For self-report measures of listening or communication difficulties in quiet and in noise on the other hand, the role of cognition is much less clear and a clear link with cognition is not always shown.[8] It is unclear why this link is so variable. Again, the cognitive ability most likely to be assessed is working memory. Maybe the listening situations most commonly assessed with self-report measures of communication are not of the type that require working memory or are more complex listening situations that would necessitate the involvement of executive functions. This idea would make sense given that listening and communicating in complex and noisy environments draw on the ability to shut out distractions and maintain focus. And thus it is conceivable that differences in *executive functions* may play a key role in the variation of individual experiences of listening and communication difficulties, regardless of absolute hearing levels.

Executive functions refer to 'higher order cognitive processes that control lower level cognitive processes in the service of goal-directed behaviour' (p.186).[9] They enable the ability to think before acting, plan, meet novel and unanticipated challenges, resist temptations and maintain focus.[10] According to Miyake and Friedman,[11] there are three core executive functions: mental-set shifting (*shifting*), information updating and monitoring (*updating*), and inhibition of prepotent responses (*inhibition*). Indeed, there is emerging evidence from large-scale cohort studies that individuals with self-reported hearing loss exhibit significantly poorer performance on tests of flexibility, psychomotor speed and executive function.[12] Similarly, a systematic review of tinnitus research found individuals who reported tinnitus had poorer performance on measure of executive function compared with individuals who did not.[13] Subsequent empirical research showed that for a population of adults with tinnitus, those reporting that their tinnitus was bothersome showed poorer

performance on measures of executive function compared with those reporting non-bothersome tinnitus.[14]

In this review, we aim to synthesise the evidence assessing the relationship between self-reported listening and communication difficulties and objective measures of executive function, while controlling (where possible) for the potentially confounding factors of age and pure-tone audiometric hearing thresholds. To our knowledge, this independent relationship has yet to be extensively examined, despite data pertaining to both executive functions and self-reported listening and communication difficulties often being reported as part of wider research study methods.

When considering measures of self-reported listening and communication difficulties, it is important to clearly define what we mean, as there are well over a hundred self-report measures pertaining to listening,[15] and only a subset will be relevant to our current research question. For this reason, we adopt definitions of listening difficulties provided by the ICF as *activity limitations* and *participation restrictions* arising from hearing loss, and narrow our focus to self-report measures that align with ICF core set for hearing loss domains of *listening* (d115), *communicating with – receiving – spoken messages* (d310) and *conversation* (d350). Similarly, to definitively identify executive function domains and classify behavioural measures of executive function as either shifting, updating or inhibition, we will use the Cattell-Horn-Carroll-Miyake (CHC-M) taxonomy.[16]

### Review questions

#### Primary research question

Is there an association between self-reported listening and communication difficulties and performance on behavioural measures of executive function in adults with hearing loss?

#### Secondary research question

Is any association moderated by age and/or hearing loss (as measured using average pure-tone audiometric thresholds)?

### Objectives

To review and synthesise evidence for the association between self-reported listening difficulties and performance on behavioural measures of executive function, in adults with hearing loss.

## METHODS AND ANALYSIS

### Eligibility criteria

#### Participants

Adults with hearing loss (with or without hearing devices), aged 18 years and over with no reported cognitive decline. We will accept a qualitative definition of hearing loss as 'mild', 'moderate', 'severe' or 'profound', or a quantitative definition where the group average pure-tone audiometric threshold is classed as mild hearing loss or greater using the WHO definition of mild (26–40 decibel [dB] hearing level [HL] inclusive); moderate (41–60 dB HL inclusive); severe (61–80 dB HL inclusive) and profound (81+dB HL).[17] Studies that report on mixed populations (eg, children and adults or normal hearing participants and participants with hearing loss) will be included only if the data for the populations of interest are reported separately.

#### Intervention/interest

A correlation coefficient between self-reported listening or communication difficulties and executive function, either reported or calculated from other reported data.

#### Outcomes

Self-reported listening and communication difficulties can be measured by a single item or a questionnaire assessing the following ICF core set for hearing loss domains of listening (d115), communicating with—receiving—spoken messages (d310) and conversation (d350).

At least one behavioural measure of executive function must be included, defined according to the CHC-M taxonomy as tasks that measure updating (eg, verbal N-back), shifting (eg, trail making part B) and inhibition (eg, Stroop).[16]

Where available, demographic information about the population (age, hearing device, group description) and objectively measured hearing loss (average pure-tone audiometric thresholds) will also be examined as subgroup descriptors and/or potential moderator(s).

#### Study design

Cross-sectional, longitudinal, experimental, quasi-experimental and observational studies will be included.

### Information sources

Articles must be available in English. No restrictions on publication dates will be applied.

Databases to be searched (see box 1 for search terms): MEDLINE (via Ovid SP), EMBASE (via Ovid SP), PsycINFO (via Ovid SP), ASSIA (via ProQuest), Cumulative Index to Nursing and Allied Health Literature or CINAHL (via EBSCO Host), Scopus, PubMed and Web of Science (Science and Social Science Citation Index).

---

**Box 1  Search terms for databases (a full search strategy is provided in the online supplemental file 1)**

**MEDLINE (OVID)** exp=explode the search term to include narrower more specific terms, .af. = search all fields in the document
1. exp Hearing Loss/
2. exp Hearing/
3. exp Self Report/
4. (self report* or self-report* or questionnaire).af.
5. exp Cognition/
6. (cogniti* or executive or attention* or memory).af.
7. (inhibit* or updat* or shift*).af.
8. 1 or 2
9. 3 or 4
10. 5 or 6 or 7
11. 8 and 9 and 10

---

Grey literature including PhD theses, unpublished datasets and conference proceedings are eligible for inclusion. Unpublished data will be accessed by contacting the corresponding authors of identified records. Literature searches were carried out on 11 May 2022.

### Article selection process

Two reviewers will independently screen titles and abstracts, and full texts of retrieved records, against the inclusion and exclusion criteria. If insufficient information is provided in the titles and abstracts to know if it should be included or if there is disagreement between the two reviewers, the article will be included in the full-text screening. Disagreement at the full-text screening will be resolved by a third reviewer.

### Data extraction process

A data extraction form will be created and improved by pilot testing before data extraction starts. The data from each study will be extracted separately by two reviewers and then compared. A third reviewer will be involved if there is any disagreement. Article selection and data extraction will be carried out using Covidence review management software (https://www.covidence.org/).

### DATA ITEMS

The data to be extracted are the aim, study design, setting, conflicts of interest, demographic information about the population (age, hearing device, group description), sample size, bibliographic information (publication year, authors, journal), correlation coefficients between self-reported listening difficulties and executive function (and (if reported) between pure-tone audiometric thresholds and self-reported listening difficulties/executive function), type of executive function measure, type of self-reported listening difficulty measure and (where relevant) procedure of pure-tone audiometric assessment, as well as documenting any missing outcome data. We will note if both self-report and behavioural measures have been completed while wearing a hearing device. The authors will be contacted via email if sufficient detail is not reported. If data are only reported via figures, then WebPlotDigitizer (http://arohatgi.info/WebPlotDigitizer/app/) will be used to extract data. A third reviewer will be involved if there is any disagreement between data extracted.

### Study risk of bias assessment

Two reviewers will assess risk of bias for each study identified by for each study using the appropriate JBI critical appraisal tool. If disagreements arise, a third reviewer will be involved. The JBI critical appraisal tools include assessment of methodological quality, and different checklists are used depending on the design of the study (eg, cross-sectional, longitudinal, randomised controlled trial). For each criteria, studies will be assessed for fulfilment (yes, no, unclear or not applicable).

### Data synthesis

Key study characteristics will be described, including study design, sample size and type of executive function measures used. The effect to be synthesised is the relationship between self-reported listening difficulties and behavioural measures of executive function defined by correlation coefficients. If correlation coefficients cannot be calculated or extracted for meta-analysis, study authors will be contacted to request the required information. Subgroup analyses will examine (where reported) key factors of age, category/measure of self-reported listening difficulty, type of executive function measure, and type of hearing device, and pure-tone audiometric thresholds. Meta-analyses will be conducted for subgroups where data for a minimum of n=5 effects are available.

The meta-analysis will be carried out using the correlation coefficient as the outcome measure. A random-effects model will be fitted to the data. We will calculate at maximum one correlation coefficient per type of executive function by type of listening difficulty. If a study reports multiple different correlations, there is likely to be some level of dependency in the data that needs to be dealt with. To handle any dependency between effect sizes in the analyses, a multilevel random-effects meta-analysis approach, as recommended by Assink and Wibbelink, will be applied.[18] This approach includes one random effect for each study as an addition to the random effect for each effect size. Likelihood ratio tests will compare the fit of the multilevel model to the fit of the reduced models. If the multilevel random-effects analysis has better fit, it will be used in all analyses, otherwise the random-effects model will be used.

The amount of heterogeneity (ie, $\tau 2$) will be estimated using the restricted maximum-likelihood estimator.[19] In addition to the estimate of $\tau 2$, the Q-test for heterogeneity and the $I^2$ statistic will be reported.[20 21] In case any amount of heterogeneity is detected (ie, $\tau\hat{}2>0$, regardless of the results of the Q-test), a prediction interval for the true outcomes will also be provided.[22] Studentized residuals and Cook's distances will be used to examine whether studies can be defined as outliers and potentially influential in the context of the model.[23] Studies with a studentized residual larger than the $100 \times (1-0.05/(2\times k))$th percentile of a standard normal distribution will be considered potential outliers (ie, using a Bonferroni correction with two-sided $\alpha=0.05$ for k studies included in the meta-analysis). Studies with a Cook's distance larger than the median plus six times the IQR of the Cook's distances will be considered influential. The analysis will be carried out using R and the metafor package.[24 25]

The analysis for the secondary research question will be carried out in the same way as above, with the difference that meta-regressions will be used to investigate potential moderator effects. The moderators will be evaluated one by one, and if both get significant effects, they will be evaluated together. The meta-regression procedure will follow the tutorial for meta-regression on the metafor package home page.[26]

## Reporting bias assessment

Funnel plots will be used to assess reporting bias. In addition, the funnel plot asymmetry will be evaluated with the rank correlation test and the regression test, using the SE of the observed outcomes as predictor.[27 28]

## Ethics and dissemination

This review does not raise any ethical issues. Results will be disseminated via scientific peer-reviewed journal articles, scientific magazines and conference presentations.

## Patient and public involvement

Patients or the public were not involved in the creation of this review protocol.

## Study design

MeSH terms will be used in relevant databases.

**Author affiliations**
[1]Psychology, Nottingham Trent University, Nottingham, UK
[2]Department of Behavioural Sciences and Learning, Linköping University, Linkoping, Sweden
[3]Manchester Centre for Audiology and Deafness (ManCAD), Division of Psychology, Communication and Human Neuroscience, The University of Manchester, Manchester, UK
[4]Hearing Sciences, Mental Health and Clinical Neurosciences, University of Nottingham School of Medicine, Nottingham, UK
[5]National Institute for Health & Care Research (NIHR), NIHR Nottingham Biomedical Research Centre, Nottingham, UK

**Contributors** HH, AH and HD developed the study. HH and AH created the search terms. JES, AH and HH wrote the review protocol and HD wrote the meta-analysis methods. EH and LB developed the data extraction plan and RoB assessment. AH, HD, EI, EH, HH and LB provided critical feedback on drafts of the protocol.

**Funding** This research is supported by the National Institute for Health and Care Research (NIHR) [CDF-2018-11-ST2-016 and PB-PG-0816-20044], the NIHR Nottingham Biomedical Research Centre [NIHR 203310], the NIHR Manchester Biomedical Research Centre [NIHR 203308], and by the Swedish Research Council [grant 2017-06092]. The views expressed are those of the author(s) and not necessarily those of the NIHR or the Department of Health and Social Care.

**Competing interests** None declared.

**Patient and public involvement** Patients and/or the public were not involved in the design, or conduct, or reporting, or dissemination plans of this research.

**Patient consent for publication** Not applicable.

**Provenance and peer review** Not commissioned; externally peer reviewed.

**ORCID iDs**
Jemaine E Stacey http://orcid.org/0000-0003-4035-712X
Helen Henshaw http://orcid.org/0000-0002-0547-4403

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
