## [Reviewer comments · BMJ Open]

ARTICLE DETAILS

TITLE (PROVISIONAL)	The relationship between self-reported listening and communication difficulties and executive function: A protocol for a systematic review and meta-analysis
AUTHORS	Stacey, Jemaine; Danielsson, Henrik; Heinrich, Antje; Batinović, Lucija; Holmer, Emil; Ingo, Elisabeth; Henshaw, Helen

VERSION 1 – REVIEW

REVIEWER	Oscar Cañete The University of Auckland, School of Psychology
REVIEW RETURNED	23-Jan-2023

GENERAL COMMENTS	It is not clear the limitations of the review, there are any?
---

REVIEWER	Moumita Choudhury Texas Tech University Health Sciences Center
REVIEW RETURNED	15-Feb-2023

GENERAL COMMENTS	This is a novel Protocol, and the researchers have taken care of data collection and analyses. No concerns here.
--

REVIEWER	Maria Hoff University of Gothenburg
REVIEW RETURNED	22-May-2023

GENERAL COMMENTS	Review for the BMJ Open of the study protocol for the study entitled The relationship between self-reported listening and communication difficulties and 2 executive function: A protocol for a systematic review and meta-analysis. Thank you for the opportunity to review this study protocol. I find that the rationale, objectives and methods for the proposed study are clearly described and well justified. The results should be of value for the research community concerned with hearing, listening and communication difficulties. The authors provide a good case for the need of this study, which will be addressing a specific gap in the knowledge on the association between listening/communication difficulties and executive function. The various outcomes that are to be studied are appropriately defined and linked to theory, such as the WHO's ICF framework. As far as I can tell, all the criteria in the provided checklist are demonstrated to have been fulfilled. I do not see any need to revise the protocol, but have some general reflections to offer that the authors might find helpful as they move forward with the proposed study: - The authors appropriately plan to control for hearing loss, by
---

	including pure-tone audiometric data. Many studies will presumably report the pure-tone average of 0.5, 1, 2 and 4 kHz. If possible, it would also be of interest to consider high frequency pure-tone averages and/or audiogram configurations. - The authors state that they will study adults with hearing loss. Is there a need to specify what degree and/or type of hearing loss, i.e. will you include studies on severe/profound hearing loss and/or conductive hearing loss? Also, will you assess left and right ears separately? My point is that hearing function involves more than audiometric sensitivity. - For subjects with hearing loss, is it possible to keep track of whether the cognitive behavioural measures were performed with or without hearing aids on? In addition, were the self-report measures on listening and communication difficulties taking hearing aid use into account? (My experience is that it is not so clear from a questionnaire if the respondent should report their difficulties with or without hearing aids). This may also be something to consider as a reason for the statement on lines 33-35: “For self-report measures of listening or communication difficulties in quiet and in noise on the other hand the role of cognition is much less clear and a clear link with cognition is not always shown [8]. It is unclear why this link is so variable”. Finally, I would like to wish the authors good luck in their pursuit of completing the study and look forward to seeing the results.
--	--

VERSION 1 – AUTHOR RESPONSE

Reviewer: 1

Dr. Oscar Cañete, The University of Auckland

Comments to the Author:

It is not clear the limitations of the review, there are any?

Author comment: Thank you for your comment, the limitations are listed in the Article Summary “Strengths and limitations” of the study. We state “only studies available in English are eligible for inclusion”

Reviewer: 2

Dr. Moumita Choudhury, Texas Tech University Health Sciences Center

Comments to the Author:

This is a novel Protocol, and the researchers have taken care of data collection and analyses. No concerns here.

Author comment: Thank you very much for reading and evaluating our manuscript.

Reviewer: 3

Dr. Maria Hoff, University of Gothenburg

Comments to the Author:

Review for the BMJ Open of the study protocol for the study entitled The relationship between self-reported listening and communication difficulties and 2 executive function: A protocol for a systematic review and meta-analysis.

Thank you for the opportunity to review this study protocol. I find that the rationale, objectives and methods for the proposed study are clearly described and well justified. The results should be of value for the research community concerned with hearing, listening and communication difficulties.

The authors provide a good case for the need of this study, which will be addressing a specific gap in the knowledge on the association between listening/communication difficulties and executive function. The various outcomes that are to be studied are appropriately defined and linked to theory, such as the WHO's ICF framework.

As far as I can tell, all the criteria in the provided checklist are demonstrated to have been fulfilled.

I do not see any need to revise the protocol, but have some general reflections to offer that the authors might find helpful as they move forward with the proposed study:

- The authors appropriately plan to control for hearing loss, by including pure-tone audiometric data. Many studies will presumably report the pure-tone average of 0.5, 1, 2 and 4 kHz. If possible, it would also be of interest to consider high frequency pure-tone averages and/or audiogram configurations.

- The authors state that they will study adults with hearing loss. Is there a need to specify what degree and/or type of hearing loss, i.e. will you include studies on severe/profound hearing loss and/or conductive hearing loss? Also, will you assess left and right ears separately? My point is that hearing function involves more than audiometric sensitivity.

- For subjects with hearing loss, is it possible to keep track of whether the cognitive behavioural measures were performed with or without hearing aids on? In addition, were the self-report measures on listening and communication difficulties taking hearing aid use into account? (My experience is that it is not so clear from a questionnaire if the respondent should report their difficulties with or without hearing aids). This may also be something to consider as a reason for the statement on lines 33-35: "For self-report measures of listening or communication difficulties in quiet and in noise on the other hand the role of cognition is much less clear and a clear link with cognition is not always shown [8]. It is unclear why this link is so variable".

Author comment: Thank you for your thoughtful comments. We have modified the data extraction section to include whether outcomes were tested aided or unaided (behavioural and self-report measures).